# Moderate Hyperkalemia Regulates Autophagy to Reduce Cerebral Ischemia-Reperfusion Injury in a CA/CPR Rat Model

**DOI:** 10.3390/brainsci13091285

**Published:** 2023-09-04

**Authors:** Xiaoqin Wang, Xinyue Tian, Haiying Shen, Xiaohua Zhang, Lu Xie, Menghua Chen

**Affiliations:** 1The Intensive Care Unit, The Second Affiliated Hospital of Guangxi Medical University, Nanning 530007, China; wxqjy8041@163.com (X.W.); xinyuetian0204@163.com (X.T.); shy98620@163.com (H.S.); 2The Department of Physiology, Guangxi Medical University, Nanning 530021, China; zhangxiaohua0217@163.com (X.Z.); xielu8282@163.com (L.X.)

**Keywords:** cardiac arrest/cardiopulmonary resuscitation, cerebral ischemia-reperfusion injury, moderate hyperkalemia, autophagy, mTOR-ULK1-Beclin1 pathway

## Abstract

Background: Cerebral ischemia-reperfusion injury (CIRI) can cause irreversible brain damage and autophagy has been implicated in the pathophysiology. Increasing serum potassium (K^+^) levels reduces CIRI, but the relationship between its protective mechanism and autophagy is unclear. In this study, we aimed to find the optimal degree of raising serum (K^+^) and to investigate the relationship between high (K^+^) and autophagy and the underlying mechanisms in a cardiac arrest/cardiopulmonary resuscitation (CA/CPR) rat model. Methods: Sprague Dawley (SD) rats were divided into four groups: S group, N group, P group, and Q group. The rats S group and N group were administered saline. The rats P group and Q group were administered 640 mg/kg of potassium chloride (KCl) continuously pumped at 4 mL/h (21.3 mg/(kg·min) and divided according to the electrocardiogram (ECG) changes during the administration of KCl. After 24-h of resuscitation, neural damage was assessed by measuring neurological deficit score (NDS), oxidative stress markers, and pathological staining of the cerebral cortex. The level of autophagy and the expression of mTOR-ULK1-Beclin1 pathway-related proteins were evaluated using transmission electron microscopy (TEM), immunostaining, and western blotting. Results: Our results revealed that high (K^+^) improved NDS and decreased the oxidative stress markers. The autophagosomes, autolysosomes, and lysosomes were decreased following treatment KCl. Furthermore, the levels of micro-tubule-associated protein 1 light chain 3 (LC3) Ⅱ/Ⅰ, Unc-51-like kinase 1 (ULK1), and Beclin1 were decreased, whereas mTOR expression was increased in the cortex. Conclusion: The results demonstrated that moderate hyperkalemia could alleviate autophagy after CIRI via regulating the mTOR-ULK1-Beclin1 pathway.

## 1. Introduction

Cardiac arrest/cardiopulmonary resuscitation (CA/CPR) patients have poor clinical outcomes with high mortality and disability rates, and cerebral ischemia-reperfusion injury (CIRI) following resuscitation is one of the main contributing factors to this outcome. CIRI is characterized by a variety of pathophysiological changes, such as cellular energy metabolism, excitatory neurotransmitter release, mitochondrial damage, oxidative stress, and inflammatory factor release, in the hours to days following resuscitation [1,2]. Collectively, these mechanisms contribute to the death of neuronal cells and poor neurological function.

Among all the forms of cell death, macroautophagy (simply as autophagy) plays a key role during the CIRI which could maintain the homeostasis of cells [3,4,5,6]. Neural cells that are characterized by high energy demands and limited energy reserves seem to be more susceptible to autophagic cell death [3]. Autophagy also has been reported in CIRI increasingly, both in global [7,8,9] and focal ischemia [10,11,12,13], and it has been demonstrated that regulating autophagy levels can mitigate CIRI. This suggests that the consideration of autophagy as a therapeutic target for CIRI is of real relevance. Our previous research has shown that moderately increasing the serum potassium (K^+^) can mitigate CIRI and enhance the neural function of rats in the MCAO model, either before or after reperfusion [14,15]. However, it is not clear whether this protective effect is achieved through the regulation of autophagy.

And before that, there is one important issue that remains unresolved. As we know, a moderate increase in serum (K^+^) is not only beneficial in reducing CIRI but also plays a significant role in reducing cardiac ischemia-reperfusion [16,17,18]. However, it is not the case that the higher the serum (K^+^) level, the more protective it is against ischemia-reperfusion injury in the heart and brain. Excessively elevated serum (K^+^) inhibits cardiac function, leading to hemodynamic disturbances and affecting the blood supply to tissues and organs, instead aggravating ischemia-reperfusion injury. Therefore, the extent of raising serum (K^+^) levels is an essential problem that remains to be addressed. As we all know, severe hyperkalemia manifests remarkable cardiotoxicity and is prominently observed on electrocardiograms (ECGs), such as severe conduction block, ventricular fibrillation, and pacing arrest [19]. The presence of these fatal arrhythmias on the electrocardiogram leads to a pronounced reduction in the heart’s pumping function, making it challenging to maintain mean arterial pressure (MAP) within physiological regulation. This severe impairment significantly affects organ perfusion.

In the present study, we increased serum (K^+^) levels in rats by intravenously administrating potassium chloride (KCl). Under this premise, observe whether the higher serum (K^+^) levels have a better protective effect for CIRI. Additionally, determine whether this protective effect is achieved via the regulation of autophagy levels and study the underlying mechanism.

## 2. Materials and Methods

### 2.1. Animal Preparation and Animal Grouping

Healthy adult male Sprague-Dawley rats (220–270 g) were provided by the Experimental Animal Centre of Guangxi Medical University. The animal care and handling during the experiment conformed to animal ethics standards and were approved by the Animal Management and Application Committee of Guangxi Medical University (NO. 202205009).

Rats were randomly allocated into 4 groups as follows: (1) Sham group (S group), which underwent surgical manipulation and received administration of 0.67 mL of normal saline with a micro-infusion pump (Shanghai Land, LD-P2020II, Shanghai, China) at a rate of 4 mL/h, but did not undergo CA/CPR; (2) Model group, also known as normal saline group (N group), which received administration of 0.67 mL of saline (4 mL/h) and underwent CA/CPR; (3) Moderate hyperkalemia P group (P group), which received continuous administration of potassium chloride (KCl, 21.3 mg/(kg·min)). Whenever the rats’ ECG indicated a reduction in P wave amplitude to half of its baseline, the administration of KCl was immediately terminated, and CA/CPR was performed. (4) Moderate hyperkalemia Q group (Q group), which received continuous administration of KCl (21.3 mg/(kg·min)). When the rats’ ECG indicated QRS complexes widening to three times their baseline value, the administration of KCl was immediately terminated, and CA/CPR was performed. 0.5ml of arterial blood was collected from a femoral arterial catheter to monitor the serum (K^+^) content before CA was performed. A total of 72 rats were used in the entire experiment. 12 rats were excluded from the study because they experienced resuscitation failure or had poor tolerance to potassium (MAP below 60 mmHg or occurrence of fatal arrhythmias before cardiac arrest was performed). Ultimately, 60 rats were successfully modeled. Each group comprised 15 rats for the purpose of analysis and detection. After 24 h of ROSC, the cerebral cortex of the rats was harvested for further experimentation. The details of experimental grouping and procedures are shown in Figure 1.

### 2.2. Study Protocol and Establishment of the CA/CPR Rat Model

After anesthetizing, the rats were secured to the operating table, and two catheters were inserted into the right femoral artery and the right femoral vein, respectively. The arterial catheter was connected to a pressure transducer (BL-420E Bio-Systems, Chengdu Technology & Market Co., Ltd., Chengdu, China) to monitor MAP, and the venous catheter was connected to a micro-infusion pump for administration of fluids. In our unpublished study, we discovered that continuous intravenous administration of 21.3 mg/(kg·min) of KCl in rats, caused a gradual reduction and eventual disappearance of P wave amplitude on ECG. This was followed by the widening of QRS complexes, fusing with T waves, and further occurring severe conduction block and/or arrhythmia, as well as a decrease in MAP to below 60 mmHg. Previous studies have demonstrated that rats are able to maintain self-regulation of cerebral blood flow within the necessary range when MAP fluctuates between 60–140 mmHg [20,21]. Accordingly, all rat groups in our experiment were carefully monitored to ensure MAP ≥ 60 mmHg before inducing CA. The P wave and QRS complex are key parameters in ECG diagnosis, known to exhibit consistent alterations when serum (K^+^) levels are elevated. For this reason, we utilized P wave amplitude reduction to half of the baseline and QRS complex widening to three times baseline as the grouping criteria to classify high potassium levels.

Before administering the NS or KCl, the MAP and baseline ECG parameters of the rats were recorded. After starting the administration, the changes in MAP, HR, and dynamic ECG of the rats were observed. Limbs were connected to subcutaneous needle electrodes, and ECGs were recorded with a Japanese photoelectric electrocardiograph (ECG-1350P, this system relies on non-digitized data, and the electrocardiograms were read by a trained professional). The observation lead was lead II. The models of CA/CPR in rats were performed as before [9,22,23]. In simple terms, CA was induced using an esophageal stimulation electrode (12V) (Chengdu Technology & Market Co., Ltd., Chengdu, China) lasting for 1 min. Six minutes post-CA, the rats received CPR, which involved intubation and assisted ventilation with a small animal ventilator (RWD LIFE Science Co., Ltd., RWD 407, Shenzhen, China) operating at a frequency of 70 breaths/min, with an oxygen concentration of 21%, a tidal volume of 6 mL/kg, and a positive end-expiratory pressure (PEEP) of 0 cmH_2_O. Mechanical chest compressions were performed at a rate of 180 compressions per minute, with a compression depth of 25–30% of the diameter of the rat’s chest, and equal compression and relaxation time intervals. One minute after starting CPR, adrenaline (20 mg·kg^−1^) was injected via the venous catheter. The successful return of spontaneous circulation (ROSC) was defined as an attained MAP > 60 mmHg for ≥1 min and the presence of a regular heart rate. If the MAP remained less than 60 mmHg after five minutes of CPR, it was classified as a failure of ROSC, and resuscitation was terminated. The rectal temperature of the rats was continuously monitored from the initial anesthesia to the recovery period, and a heating lamp was used to sustain the temperature at 37.0 ± 0.5 °C. Once the rats had fully recovered, they were relocated to their cages and housed in a peaceful, air-conditioned room at a constant temperature of 24 °C.

### 2.3. Neurological Evaluation

After 24 h of ROSC, rats underwent neurological function assessments through the application of the Neurological Deficit Score (NDS) [24], which measures different criteria, including consciousness level and breathing pattern (a total score of 19 points), brain-stem function (a total score of 21 points), motor assessment (a total score of 6 points), sensory function(a total score of 6 points), motor behavior (a total score of 6 points), behavior (a total score of 12 points) and seizures (a total score of 10 points). The neurological deficit score ranges from 0 (indicating death or brain death) to 80 (indicating no observed neurological deficit). For specific scoring details, please refer to the Appendix A. Two researchers, with no knowledge of the experimental design and were well-trained, conducted the NDS evaluation and reached a consensus on the score. After the assessment, rats were euthanized, and brain tissue was harvested for additional morphological and biochemical index measurements.

### 2.4. Preparation of Brain Tissues

After 24 h of ROSC, rats were administered 2% pentobarbital sodium (90 mg/kg) to induce unconsciousness. The entire cerebral cortex tissue from 3 rats in each group was used for reactive oxygen species (ROS) detection. Each group of 3 animals was used for malondialdehyde (MDA), superoxide dismutase (SOD), and glutathione (GSH) detection. Each group of 3 animals was subjected to both morphological and pathological observations. The remaining entire cerebral cortex tissue samples from the other 6 rats in each group were stored immediately at −80 °C for western blotting.

### 2.5. Biochemical Analysis: Serum (K^+^), ROS, MDA, SOD and GSH Levels

Serum (K^+^) content was determined using a Potassium Kit (Nanjing Jian Cheng Technology Co., Ltd., Nanjing, China). The ROS assay kit (E004-1-1, Nanjing Jian Cheng Technology Co., Ltd., Nanjing, China) was used. The rat cerebral cortex was isolated and rinsed with pre-cooled PBS. The tissue was then prepared into a single-cell suspension using mesh filtration (300-mesh nylon mesh), and a fluorescent probe was added before incubating at 37 °C for 30 min. After incubation, fluorescence intensity was measured using a fluorescence plate reader (excitation wavelength of 488 nm and emission wavelength of 525 nm).

To determine brain MDA levels, SOD activity, and GSH levels, brain tissues were collected at 24 h after reperfusion and homogenized. Brain tissue homogenates were centrifuged at 1000× *g* at 4 °C for 15 min. The supernatants were then used for the measurement of MDA levels, SOD activity, and GSH levels with a MDA Assay Kit, SOD Assay Kit, and GSH Assay Kit, respectively (Nanjing Jian Cheng Technology Co., Ltd., Nanjing, China). Tissue protein was quantified using the BCA Protein Assay Kit (Boster Biological Technology Co., Ltd., Wuhan, China), and MDA levels, SOD activity, and GSH levels were calculated relative to protein concentration (*n* = 3).

### 2.6. HE and Nissl Staining

For HE staining, the paraffin-embedded tissue sections were initially deparaffinized using trichloroethylene, followed by washing with graded alcohol. Subsequently, the sections were stained with hematoxylin for 3 min and treated with 1% acid alcohol for 5 min. Once washed with tap water until they turned blue, the HE-stained tissue sections were observed under a microscope.

For Nissl staining, paraffin removal was first carried out using trichloroethylene, followed by staining the sections with the Toluidine blue solution (G3668; Solar bio, Beijing, China). Subsequently, the sections were stained with 0.5% eosin for 3 s and washed in running water. The identification of Nissl bodies was accomplished through light microscopy, and ImageJ software (National Institutes of Health, Bethesda, MD, USA) was utilized for their counting from five random sections within each of the five different regions in each group.

### 2.7. Transmission Electron Microscopy

The rats were subjected to transcardial perfusion using saline and 4% paraformaldehyde followed by fixation in 2.5% glutaraldehyde. After that, the brain was removed quickly from the rats, and 1 mm × 1 mm × 1 mm tissue blocks were excised from the right cerebral cortex. The tissue blocks were then fixed again in 2.5% glutaraldehyde, dehydrated, embedded, and sectioned using a Leica EM UC7 ultramicrotome into 80-nm sections. The ultrathin sections were then stained with uranium-citrate and examined using an electron microscope (H-7650; Hitachi, Tokyo, Japan).

### 2.8. Immunofluorescence Staining

The brain tissue cryosections were routinely thawed and washed. Then, the primary antibody LC3 (GB11124, 1:200, Service bio, Wuhan, China), and LAMP-1 (GB112949, 1:200, Service bio, Wuhan, China) were incubated overnight at 4 °C. After the second day, the tissues were washed with phosphate buffer saline (PBS) three times for 5 min, secondary antibody (GB21303, 1:300, Service bio, Wuhan, China) was incubated in a cassette for 50 min at 37 °C, and then washed with PBS again. Subsequently, cell nuclei were counterstained with 4′-6-diamino-2-phenylindole (DAPI) followed by observation and photography. The captured images were viewed with the Upright Fluorescence Stereomicroscope (Nikon Eclipse C1, Nikon, Tokyo, Japan). Double-stained cells from the different groups were counted throughout five random lesion regions in the cortex.

### 2.9. Western Blot (WB) Analysis

The protein was extracted from frozen brain tissue using RIPA lysate. The BCA method was used to measure protein concentration. The protein solution and SDS-PAGE loading buffer (5× were mixed at 1:4, then heated at 100 °C for 10 min to denature the protein. After this production, equal amounts of proteins were separated using SDS-PAGE gel and transferred onto the PVDF membrane. Then, PVDF membranes were blocked with 10% skim milk and incubated with anti-LC3 antibody (1:1000, CST, Danvers, MA, USA), anti-P62 antibody (1:1000, CST, Danvers, MA, USA), anti-mTOR antibody (1:1000, CST, USA), anti-ULK1 antibody (1:1000, CST, Danvers, MA, USA), anti-Beclin1 antibody (1:1000, CST, Danvers, MA, USA), and anti-β-actin antibody (1:1000, Service bio, Wuhan, China) overnight at 4 °C. Afterwards, the bolts were incubated for 2 h at 37 °C using the anti-rabbit IgG conjugated secondary antibodies (1:20,000, CST, Danvers, MA, USA). Finally, the blot bands were quantified using an Odyssey imaging system (LI-COR, Lincoln, NE, USA). Protein levels were normalized to β-actin as a reference. The relative density of the protein level was quantitated by the ImageJ software. All data were obtained from five independent experiments.

### 2.10. Statistical Analysis

The experimental data were processed by SPSS version 23.0 software. The data was presented as mean values ± standard deviation (SD). All data were tested for normality and homogeneity of variance. Normal data were analyzed using one-way ANOVA, while homogeneity of variance was assessed using LSD for multiple comparisons. For data with unequal variance, the Dunnett T3 test was employed for multiple comparisons. Statistical significance was considered to be *p* < 0.05. Graph layouts were achieved with GraphPad Prism8 (GraphPad Software, Inc., San Diego, CA, USA).

## 3. Results

At first, we measured baseline parameters before starting the KCl administration in the rats. It is important to note that there were no significant differences observed in this pre-administration data among the experimental groups. Then, we recorded several parameters before CA, following the discontinuation of KCL administration. These parameters included the duration and total amount of administered potassium, serum (K^+^) level, HR, and MAP. Finally, 24 h after the rats were successfully resuscitated, we recorded their body weights and NDS. Then, the cerebral cortical tissues were collected for further testing.

### 3.1. Baseline Characteristics

The baseline parameters were shown in Table 1, there were no significant differences in baseline characteristics before KCl administration among all groups.

After 24 h post-ROSC, there was a slight decrease in the body weight of the rats. Weight loss may affect neurological assessments and various molecular markers. Therefore, we recorded the body weight of each group of rats before euthanasia. There was no significant difference in body weight among the groups of rats at 24 h after resuscitation.

### 3.2. Elevating the Pre-CA Serum (K^+^) Levels in Rats Improves Neurological Function and Reduces ROSC-Induced Neuronal Damage

During the continuous administration of KCl, the administration was immediately stopped upon fulfilling the grouping requirement. Prior to CA, blood samples were collected via the femoral vein from each group to monitor pre-CA serum (K^+^) levels. The heart rate and MAP were also recorded at that time. After 8.9 ± 0.5 min of administration, the P wave amplitude in group P decreased to half of the baseline value. Simultaneously, the total amount of KCl administrated reached 43.58 ± 3.88 mg, while the serum (K^+^) level was measured to be 8.76 ± 0.45 mmol/L (Table 2). The QRS wave in group Q widened to three times the baseline value at 13.04 ± 1.04 min and the serum (K^+^) level reached 10.18 ± 0.30 mmol/L after 63.89 ± 6.66 mg of KCl was administrated (Table 2). Serum (K^+^) levels were significantly increased in groups P and Q compared to groups N before CA (*p* > 0.05). Additionally, the Q group exhibited significantly higher serum (K^+^) levels than the P group (*p* < 0.001), (Table 2 and Figure 2a). Before the induction of CA, there was no significant difference in the HR among all groups (*p* > 0.05), (Table 2 and Figure 2b). The MAP was significantly reduced in the Q group compared to the N and S groups before CA induction (*p* < 0.01). However, the Q group maintained the MAP above 60 mmHg (Table 2 and Figure 2c).

After 24 h of resuscitation, the NDS score of the N group was significantly decreased compared with the S group (*p* < 0.001). However, after appropriate elevations in serum (K^+^) levels before CA induction, the NDS scores of the P and Q groups were significantly increased compared with the N group (*p* < 0.001). Moreover, the NDS score of the Q group was significantly higher than that of the P group (*p* < 0.05), (Figure 2d).

Oxidative stress is an important mechanism of CIRI [25]. Therefore, we assessed the levels of ROS, MDA, SOD activity, and GSH levels in the cerebral cortex of all rat groups. We measured the fluorescence intensity of ROS using the DCFH-DA probe. Our findings revealed that ROS fluorescence intensity was significantly higher in the N group compared to the S group (*p* < 0.01); Interestingly, increasing serum potassium levels pre-CA induction reduced ROS generation, in particular, in the Q group, where ROS generation was significantly lower than in the N group (*p* < 0.05), (Figure 2e). The MDA detection kit results indicated a significant increase in MDA levels in the N group compared to the S group (*p* < 0.001); In contrast, MDA levels were significantly decreased in the Q group when compared to the N and P groups (*p* < 0.01 and *p* < 0.05, respectively), (Figure 2f).

We used detection kits to measure the content of antioxidant factors. Our results revealed that compared to the S group, both SOD activity and GSH levels were significantly decreased in the N group (SOD, *p* < 0.05; GSH, *p* < 0.001). Conversely, both SOD activity and GSH levels were significantly higher in the P group compared to the N group (*p* < 0.01 for both). Additionally, the Q group with higher serum (K^+^) levels exhibited even higher SOD activity and GSH levels compared to the P group (SOD, *p* < 0.05; GSH, *p* < 0.001), as depicted in Figure 2g,h.

### 3.3. HE and Nissl Staining

Elevating serum (K^+^) levels before CA induction can improve neurological function in rats after 24 h of ROSC. To examine changes in neuronal morphology among the groups, we performed HE staining and Nissl staining on the brains of each group.

HE staining revealed that neurons in the S group displayed round, intact structures and normal fiber structures. In contrast, neurons in the N group exhibited loosely arranged structures, along with necrotic cells, nuclear deformation and shrinkage, cell swelling, and vacuoles. Comparatively, both the P and Q groups, with elevated serum (K^+^) levels before CA induction, demonstrated significant improvement in normal neuronal morphology compared to the N group, and only a few necrotic neurons were observed in the Q group (Figure 3a).

We observed numerous light blue Nissl bodies in the cytoplasm of neurons in the S group. The N group exhibited significantly fewer Nissl bodies compared with the S group (*p* < 0.001), while both the P and Q groups, with elevated serum (K^+^) levels before CA induction, demonstrated a significant increase in the number of Nissl bodies (*p* < 0.001). In addition, the Q group exhibited a higher number of Nissl bodies than the P group (*p* < 0.01) (Figure 3b,c).

### 3.4. Scanning Electron Microscopy Was Performed on the Cerebral Cortex of Each Rat Group after 24 h of Reperfusion

Electron microscopy is a widely applied method to detect autophagy. To further examine the impact of elevated serum (K^+^) levels on autophagy in CIRI, we utilize transmission electron microscopy to observe cortical neuron ultrastructural changes in each group of rats following 24 h of resuscitation.

The cortical neurons in the brains of rats belonging to the S group exhibited relatively normal features. Their cytoplasm showed mild edema and there was no significant expansion in the perinuclear gap of the nucleus, which had a clear nuclear membrane. The mitochondria displayed a few cases of mild swelling with uniform matrix and arranged cristae. In some instances, moderate swelling of the mitochondria was observed along with a shallow matrix. Autophagosomes were also present in the cytoplasm, indicating the late stage of autophagy with a myelin-like appearance.

Compared to the S group, the cortical neurons in the brains of rats belonging to the N group exhibited remarkable aberrations. The cytoplasm displayed severe edema, and a significant proportion of mitochondria displayed mild to moderate swelling, an elliptical shape, and reduced electron density of the matrix. Only a small subset of the mitochondria displayed structured cristae. The N group demonstrated abundant vacuoles, dark lysosomes, and autophagosomes present in the cytoplasm. Compared to the N group, the cortical neurons in the brains of rats assigned to the P and Q groups showed diminished damage, exhibiting partial fusion and mildly to moderately swollen elliptical-shaped mitochondria, accompanied by shallow matrix and partially structured cristae. Only a few cristae breaks were observed. A small number of autophagic lysosomes and autophagosomes were present. Figure 4 displays the electron microscopy findings of cortical neurons in the brains of rats in each group.

### 3.5. Expression of Autophagosome Membrane Protein LC3 and Autophagy Adaptor P62 in Cerebral Cortex of Rats

Microtubule-associated protein 1 light chain 3 (LC3) localizes to autophagosomes, autolysosomes, and autophagic vacuoles. The expression of LC3 was observed 24 h after ROSC via immunofluorescence. Immunofluorescence analysis revealed a significant increase in LC3 expression in the N group compared to the S group, with a marked rise in red fluorescence (*p* < 0.01); Conversely, the red fluorescence of the P and Q groups showed a significant reduction compared to the N group, with the Q group exhibiting the most significant decrease (*p* < 0.01); Notably, the red fluorescence intensity was relatively weaker in the Q group than in the P group (*p* < 0.05) (Figure 5a,b).

Western blot was conducted to detect the expression of LC3Ⅱ/LC3Ⅰ and sequestosome (SQSTM1/P62) protein, an autophagy adaptor, in each group. The immunoblot analysis of LC3Ⅱ/LC3Ⅰ demonstrated a notable increase in the LC3Ⅱ/LC3Ⅰ ratio in the N group compared to the S group (*p* < 0.001). Conversely, the LC3Ⅱ/LC3Ⅰ ratio exhibited a pronounced decrease in both the P and Q groups compared to the N group (*p* < 0.01). The LC3Ⅱ/LC3Ⅰ ratio in the Q group was notably lower than in the P group as well (*p* < 0.01) (Figure 5c,d). We observed a significant decrease in the expression of P62 protein in the N group compared to the S group (*p* < 0.01) and a notable increase in the expression of P62 protein in the Q group compared to both the N and P groups (*p* < 0.05), as revealed by the immunoblot analysis (Figure 5c,e). This implies that the increased serum (K^+^) level can influence the autophagy level after CIRI.

### 3.6. Lysosome Expression and Co-Expression with LC3 in Cerebral Cortex of CA/CPR Rats

Following autophagosome formation, transportation to lysosomes for fusion is necessary to degrade enclosed cargoes into small molecules for recycling [26,27]. Therefore, to monitor autophagic levels, changes in both LC3 and lysosomes should be tracked. We employed immunofluorescence to examine the expression of the lysosomal marker lysosome-associated membrane protein-1 (LAMP-1) and its co-expression with LC3.

The results demonstrated a significantly higher ratio of LAMP-1 positive cells/DAPI in the N group compared to the S group (*p* < 0.001), whereas the ratio in the Q group was significantly decreased compared to the N group (*p* < 0.05) (Figure 6a,b). Additionally, a significantly higher ratio of double-stained LC3 + LAMP-1 cells/DAPI was observed in the N group versus the S group (*p* < 0.001). Conversely, the ratio of double-stained LC3 + LAMP-1 cells/DAPI was significantly lower in the P and Q groups than in the N group (*p* < 0.05) (Figure 6a,c). Our results indicate that after CIRI, the expression level of lysosomes in the cerebral cortex of rats increases, along with a significant increase in the formation of autophagic lysosomes. Moreover, increasing serum (K^+^) levels reduce the generation of both lysosomes and autophagic lysosomes.

### 3.7. The Effect of Moderately Increasing Serum (K^+^) Levels on the mTOR-ULK1-Beclin1 Pathway

The results of our experiment indicate that a moderate increase in serum (K^+^) levels can regulate autophagy and improve brain function in CA/CPR rats. Previous studies have shown that the mTOR plays a crucial role in autophagy induced by CIRI [28]. As an upstream signal, mTOR regulates downstream ULK1 and Beclin1, thereby modulating the initiation of autophagy [29]. Therefore, we investigated the expression of relevant proteins in the mTOR-ULK1-Beclin1 signaling pathway.

The expression of mTOR, ULK1, and Beclin1 in the brain tissues was measured using western blot analysis. As shown in Figure 7a–f, the expression of mTOR in the N group was sharply lower than that in the S group (*p* < 0.01), and the protein expressions in the Q group were higher than in the N group and P group (*p* < 0.01). The expression of ULK1 and Beclin1 in the N group was sharply higher than those of protein expressions in the S group (ULK1, *p* < 0.01; Beclin1, *p* < 0.05), and those protein expressions in the Q group were sharply lower than that in the N group and P group (*p* < 0.05). Reductions in mTOR expression led to activation of ULK1 and Beclin1, which promote autophagy. Modest increases in serum (K^+^) levels can boost mTOR expression, reduce ULK1 and Beclin1 expression, and result in lowered autophagy levels.

## 4. Discussion

In this study, the results showed that raising serum (K^+^) levels led to decreased neurological dysfunction, improved neuronal morphology, reduced formation of autophagosomes and lysosomes, and lowered the LC3Ⅱ/LC3Ⅰ ratio in CA/CPR rats 24 h after resuscitation. Moreover, higher serum (K^+^) levels are more protective, given that the heart is able to maintain circulatory function. These effects may be attributed to the regulation of autophagy levels following cerebral ischemia-reperfusion through the mTOR-ULK1-Beclin1 signaling pathway.

CIRI results in disturbed cellular energy metabolism, depletion of ATP, damage to mitochondria, a significant increase in ROS production, calcium overload, increased conditions of excitotoxicity, and a reduction in ATPase (i.e., Na^+^-K^+^-ATPase) activity eventually culminates in nerve cell death [25]. The occurrence of those pathological processes can regulate the level of autophagy after CIRI by affecting various autophagy regulatory factors. Among the factors in mammals, mTOR acts as an energy sensor that can respond to changes in nutrient and energy levels, and further modulates the initiation phase of autophagosome formation [28,30]. During situations of energy deprivation, mTOR activity diminishes, which causes the dephosphorylation and dissociation of ULK1 from the mTOR complex [31]. After this, ULK1 interacts with Beclin1 through modifications and drives the formation of the phagophore, the initial autophagosomal precursor membrane structure [32,33]. In the presence of ROS production, autophagy is triggered by upstream activators such as AMP-activated protein kinase (AMPK), or by increased activity of autophagy-associated genes (ATG proteins) [34]. In addition, calcium overload, ATPase activity, and the condition of excitotoxicity all induce autophagy, leading to increased levels of autophagy [35,36]. This suggests that autophagy, as an energy-dependent mode of cell death, plays an important role in CIRI. Regulation of autophagy levels after CIRI may be a potential neuroprotective target. S. Xu et al. confirmed that calycosin can mitigate CIRI by suppressing the STAT3/FOXO3a signaling pathway and inhibiting overactivated autophagy, as demonstrated in both in vivo and in vitro (PC12 cells) studies [37]. Similarly, W.Y. Wang et al. have demonstrated that autophagy exacerbates CIRI while inhibiting autophagy can alleviate CIRI in their study [8]. However, some studies suggested that autophagy activation can mitigate CIRI and protect neural cells [38,39]. Possible reasons for the seemingly contradictory results may be related to the experimental models each study employed. Different levels of cerebral blood flow reduction, ischemia duration, reperfusion duration, and time elapsed after reperfusion in different models and experimental conditions can result in disparate levels of functional impairments, cell damage, and cell death [25]. Although there is no consensus on the exact effects of autophagy activation on CIRI, it is apparent that excessive or insufficient autophagy does not promote the recovery of neurological function after CIRI. Hence, it can be hypothesized that an optimal level of autophagy exists for the body. Regulating the autophagy level to this optimal level may be beneficial in both physiological and pathological states. Nevertheless, the exact optimal level remains to be extensively explored.

Similar to autophagy, serum (K^+^) also plays a critical role in maintaining the body’s homeostasis. Our previous research has shown that augmenting serum (K^+^) levels can mitigate CIRI in rats via mechanisms such as alleviating mitochondrial damage, reducing calcium overload, inhibiting the decline in Na^+^-K^+^-ATPase activity, and curtailing apoptosis, which can enhance neurological function in rats [14,15]. Considering the mechanism of autophagy activation after CIRI, we hypothesized that regulating serum (K^+^) concentration could affect the autophagy level and improve CIRI outcomes. Cellular membrane structure and function depend on the proper intake of serum (K^+^), which relies upon specific (K^+^) channels [40]. Hence, modulating serum (K^+^) levels may impact autophagy incidence by influencing (K^+^) channels, particularly ATP-sensitive potassium channels (K_ATP)_, which are associated with ATP sensitivity and energy generation. Although we did not explore this aspect in greater depth in the study. However, other research has indicated that K_ATP_ channel activation can stimulate the mTOR-autophagy regulatory pathway, leading to lower mTOR expression and stimulated autophagy [41,42,43,44]. Theoretically, elevating serum (K^+)^ levels could impede the activity of K_ATP_ channels to suppress autophagy through the mTOR signaling pathway. This conclusion coincides with the results that we obtained. We discovered that elevating the serum (K^+^) concentration of rats before CA upregulated mTOR expression, downregulated ULK1 and Beclin1 expression, and decreased autophagy level, alleviated CIRI, and improved brain function in the rats. Furthermore, autophagy serves as not only a critical mechanism for maintaining organismal homeostasis but also a specialized form of cell death termed autosis. Autosis necessitates the participation of autophagy genes as well as the Na^+^-K^+^-ATPase enzyme. Studies have revealed that the Na^+^-K^+^-ATPase and the autophagy protein Beclin1 interact with each other during stress [45]. When exposed to stressors such as starvation and ischemia-reperfusion injury in the brain and kidneys, researchers discovered an elevated interaction between Na^+^-K^+^-ATPase and Beclin1. Yet, administering Na^+^-K^+^-ATPase inhibitors resulted in a weakened interaction and reduced autophagy, leading to superior outcomes in the experimental animals [45]. The authors contend that this represents a cross-talk between the energy-generating mechanism (Na^+^-K^+^-ATPase) and the energy-utilizing mechanism (autophagy) that maintains homeostasis. We concur with the authors that regulating autophagy levels after CIRI involves regulating autophagy homeostasis. In this study, we have yet to determine whether increasing serum (K^+^) levels affect autophagy levels by modulating the interplay between Na^+^-K^+^-ATPase and Beclin1. However, this represents a promising target for us to pursue more comprehensive investigations in the future.

Undoubtedly, the aim of this study is not only to prove high serum (K^+^) levels could mitigate CIRI through inhibiting autophagy levels but also to identify an optimal concentration of high serum (K^+^) levels via the changes in ECG. Our prior study demonstrated that increasing serum (K^+^) levels can alleviate CIRI and enhance neurological function in rats affected by focal cerebral ischemia [14,15]. However, the optimal levels of high serum (K^+^) and the standard of optimal levels are undefined. Potassium (K^+^) is among the most abundant cations in cells, which can cause alterations in the resting membrane potential, nerve and muscle excitability, and cardiac electrical activity [46]. Elevated serum (K^+^) leads to changes in the electrical activity of the heart, which are reflected in the ECG and changes in the ECG can provide crucial insights into myocardial function in turn [47]. For this reason, we selected to study the optimal serum (K^+^) concentration through monitoring the changes in the rat’s ECG. As serum (K^+^) levels gradually increased, the human’s ECG first demonstrates “high and sharp” T-waves resulting from global action potential duration (APD) shortening causing more synchronous repolarization across the ventricular wall [48]. Subsequently, the P wave broadens and decreases in amplitude, eventually disappears, and the QRS widens due to conduction velocity (CV) slowing [48]. As serum (K^+^) levels continue to increase, it can lead to heart block, asystole and ventricular tachycardia (VT)/ventricular fibrillation (VF), and a significant decrease in MAP. In contrast to humans, the rats failed to observe significant “high and sharp” T-waves on the ECG during the increased serum (K^+^) levels but were able to observe complete changes in the P waves and QRS complexes [46]. Therefore, we selected the P and QRS waves with significant changes as the grouping criteria: a decrease in P wave amplitude to 1/2 the basal value was used as the low-dose group (P group), and a widening of the QRS complex to three times the basal value was used as the high-dose group (Q group). Our study results showed that there was no difference in heart rate between the low-dose group and the high-dose group before CA. Although the MAP of the high-dose group (Q group) decreased significantly compared to the N group, it remained over 60 mmHg, which also ensured cerebral blood flow to the rats [20,21].

Although the human ECG does not behave the same as the rat ECG in the presence of elevated blood potassium, our study also provides some particular insights into clinical practice. Is there a specific clinical threshold of serum potassium levels that ensures the avoidance of fatal arrhythmias and circulatory perfusion deficits while simultaneously optimizing cerebral nerve protection? Additionally, is it feasible to employ electrocardiography or other non-invasive diagnostic modalities to determine the optimal serum potassium concentration for human subjects? Of course, given the complexity of the clinical patient’s disease, this will probably be a long process. Undoubtedly, this will require significant effort to validate.

Of course, there are limitations to our study. Increasing the serum potassium level increases the efficacy of Na^+^-K^+^-ATPase and inhibits calcium overload, thereby suppressing the over-activated autophagy. However, the changes in Na^+^-K^+^-ATPase as well as calcium channels in rats before and after CA were not detected in this experiment, and the mechanism by which blood potassium regulates autophagy levels needs to be further investigated. Secondly, this study only examined macroautophagy and did not examine the various types of autophagy, such as mitophagy. Mitophagy also plays an important role in CIRI. We intend to include this aspect in further research for more in-depth investigations.

## 5. Conclusions

This study utilized a continuous intravenous infusion of potassium chloride (KCl) to elevate (K^+^) levels in a rat population. Subsequently, by analyzing changes in the electrocardiogram (ECG), the appropriate time intervals for conducting CA/CPR were determined. The results conclusively showed that elevated levels of serum (K^+^) confer greater protection against cerebral ischemia-reperfusion injury (CIRI) when the heart sustains continuous circulatory function. This protective effect can be ascribed to the tight regulation of the mTOR-ULK1-Beclin1 pathway, which efficiently modulates autophagic activity following cerebral ischemia-reperfusion injury (CIRI).

## Figures and Tables

**Figure 1 brainsci-13-01285-f001:**
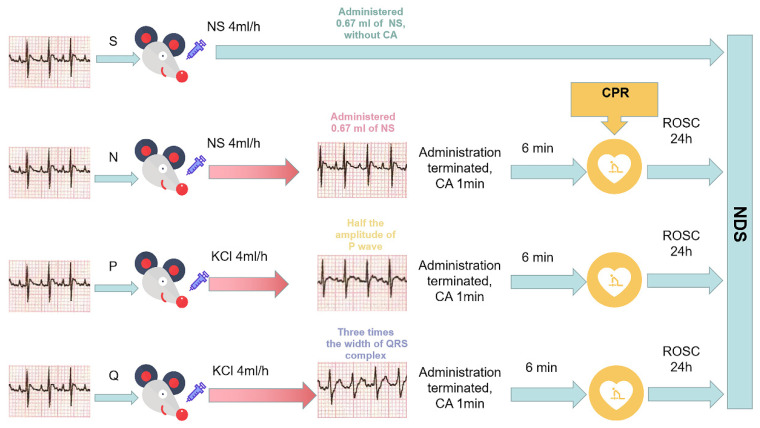
Grouping and experimental flowchart for each group. NS: normal saline; KCl:21.3 mg/(kg·min) of potassium chloride; CA: cardiac arrest; CPR: cardiopulmonary resuscitation; ROSC: return of spontaneous circulation; S: sham, normal saline; N: normal saline + CA/CPR; P: moderate hyperkalemia P + CA/CPR; Q: moderate hyperkalemia Q + CA/CPR; NDS: neurological deficit score.

**Figure 2 brainsci-13-01285-f002:**
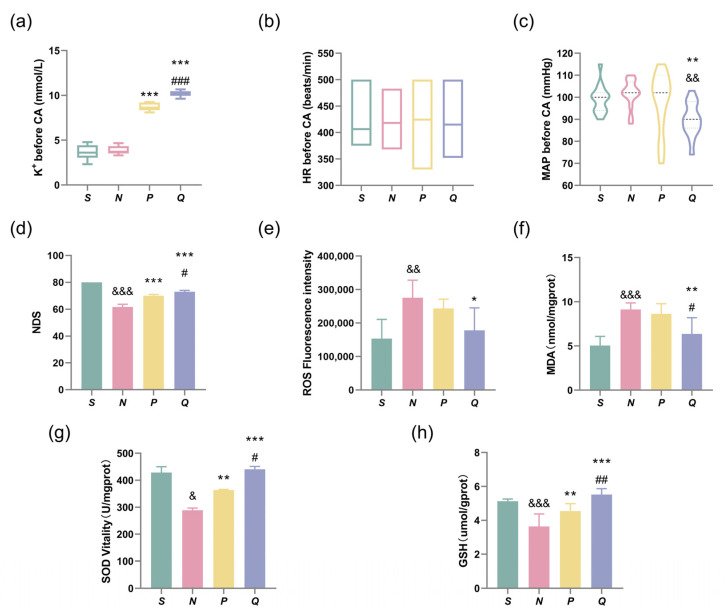
Serum (K^+^) levels, HR, and MAP before CA; neurological deficit score (NDS) and oxidative stress markers associated with brain injury at 24 h after ROSC. (**a**–**c**) Serum (K^+^), HR and MAP of each group of rats before CA were measured; (**d**) NDS of each group of rats were assessed at 24 h after CA/CPR; (**e**) ROS Fluorescence intensity of each group; (**f**) The content of MDA in each group of rats; (**g**) The results of SOD activity assay in the cerebral cortex of each group of rats; (**h**) The level of GSH in the cerebral cortex of each group of rats. All data are presented as the Mean ± SD. ^&^ *p* < 0.05, ^&&^ *p* < 0.01 and ^&&&^ *p* < 0.001 vs. the S group; * *p* < 0.05, ** *p* < 0.01 and *** *p* < 0.001 vs. the N group; ^#^ *p* < 0.05, ^##^ *p* < 0.01 and ^###^ *p* < 0.001 vs. the P group; S: sham, normal saline; N: normal saline + CA/CPR; P: moderate hyperkalemia P + CA/CPR; Q: moderate hyperkalemia Q + CA/CPR.

**Figure 3 brainsci-13-01285-f003:**
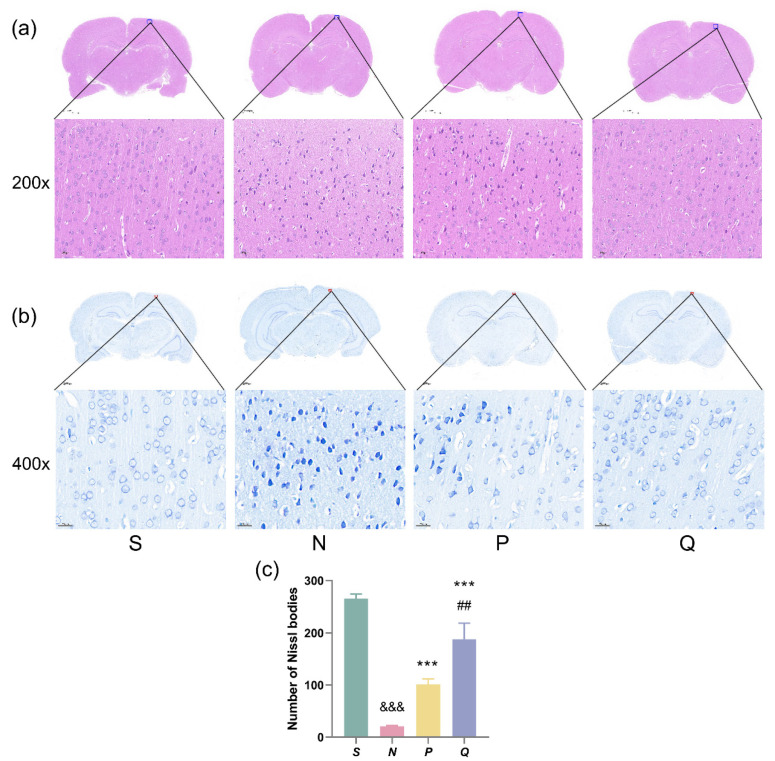
HE staining and Nissl staining of the cerebral cortex in rats. (**a**) HE staining of the cerebral cortex after ROSC for 24 h. Scale bar: 50 μm (200×); (**b**) Nissl staining of the cerebral cortex after ROSC for 24 h showing Nissl bodies. Scale bar: 20 μm (400×); (**c**) Number of Nissl bodies per group. All data are presented as the Mean ± SD. ^&&&^ *p* < 0.001 vs. the S group; *** *p* < 0.001 vs. the N group; ^##^ *p* < 0.01 vs. the P group; HE: haematoxylin and eosin; S: sham, normal saline; N: normal saline + CA/CPR; P: moderate hyperkalemia P + CA/CPR; Q: moderate hyperkalemia Q + CA/CPR.

**Figure 4 brainsci-13-01285-f004:**
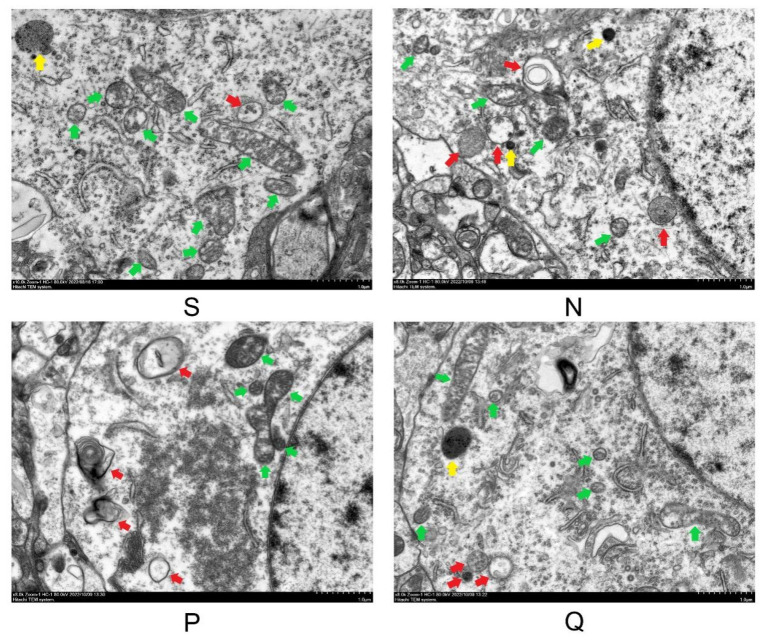
Sample transmission electron microscopy (TEM) images of cerebral cortex neurons at 24 h after ROSC. Neuronal cells exhibited a dilated endoplasmic reticulum, swollen and balloon-like mitochondria, mitochondrial crests that had fused or disappeared, and the formation of numerous vacuoles in the cytoplasm and numerous darkened lysosomes and autophagosomes. Scale bars, 1 µm (8K× x). Green arrows indicate mitochondria; yellow arrows indicate lysosomes: red arrows indicate autophagosomes and autolysosomes. S: sham, normal saline; N: normal saline + CA/CPR; P: moderate hyperkalemia P + CA/CPR; Q: moderate hyperkalemia Q + CA/CPR.

**Figure 5 brainsci-13-01285-f005:**
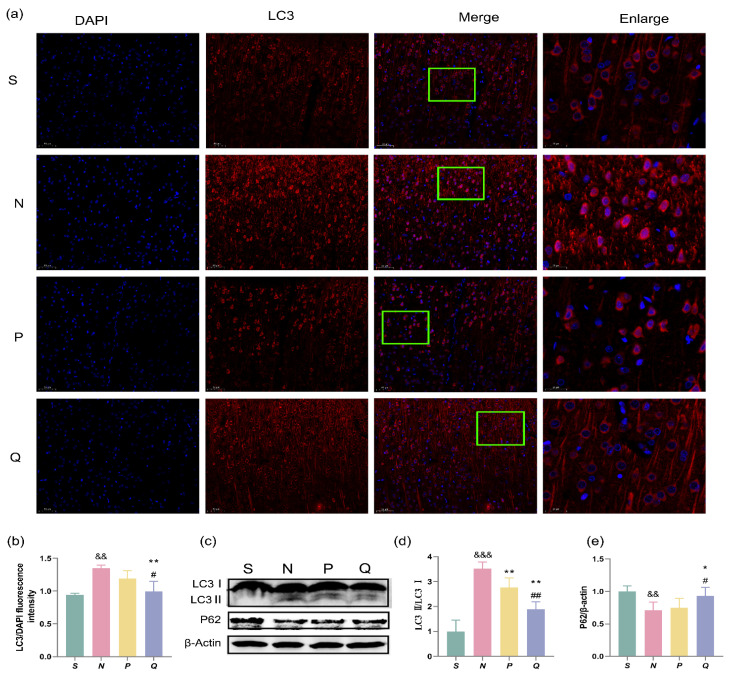
An increase in serum (K^+^) levels appropriately reduces autophagic levels in CA/CPR rats at 24 h post-resuscitation. (**a**) Representative images of LC3 immunostaining in our experimental settings. Scale bar = 50 μm (200×), Enlarge: Scale bar = 20 μm (630×); (**b**) The ratio of LC3 fluorescence value to DAPI (cellular nuclear fluorescence value), *n* = 3; (**c**) WB band of LC3 and P62; (**d**), (**e**) WB analysis of LC3 and P62. *n* = 5. Data are reported as the Means ± SD. ^&&^ *p* < 0.01 and ^&&&^ *p* < 0.001 vs. the S group; ** *p* < 0.01 and * *p* < 0.05 vs. the N group; ^##^ *p* < 0.01 and ^#^ *p* < 0.05 vs. the P group; S: sham, normal saline; N: normal saline + CA/CPR; P: moderate hyperkalemia P + CA/CPR; Q: moderate hyperkalemia Q + CA/CPR.

**Figure 6 brainsci-13-01285-f006:**
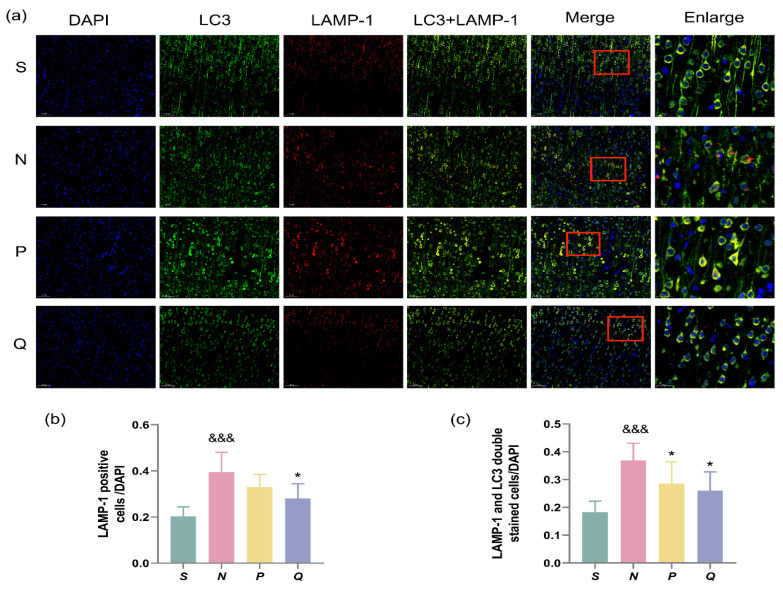
Autolysosome formation at different groups after reperfusion. LC3 (Green) and LAMP-1 (Red) co-localization indicated autolysosomes. (**a**) Representative images of autolysosome co-labelling in different groups. Scale bar = 50 μm (200×), Enlarge: Scale bar = 20 μm (630×); (**b**) The ratio of LAMP-1 positive cells and DAPI; (**c**) Ratio of LAMP-1 and LC3 double stained cells to DAPI in different groups. Data are reported as the means ± SD, *n* = 3; ^&&&^ *p* < 0.001 vs. the S group; * *p* < 0.05 vs. the N group; S: sham, normal saline; N: normal saline + CA/CPR; P: moderate hyperkalemia P + CA/CPR; Q: moderate hyperkalemia Q + CA/CPR.

**Figure 7 brainsci-13-01285-f007:**
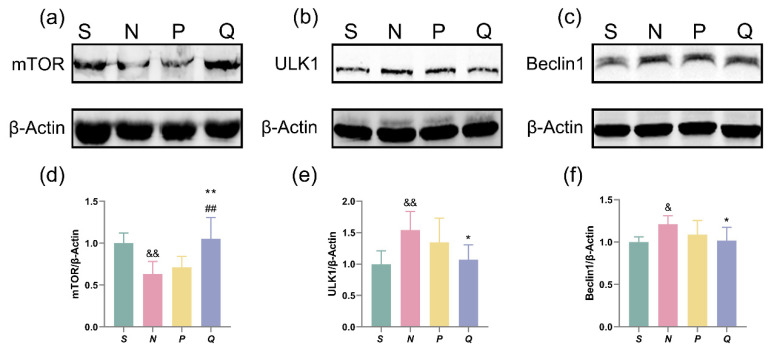
WB analysis of mTOR, ULK1, and Beclin1. (**a**–**c**) WB band of mTOR, ULK1, and Beclin1; (**d**–**f**) WB analysis of mTOR, ULK1, and Beclin1. *n* = 5. All data above are represented as the mean ± SD. ^&&^ *p* < 0.01 and ^&^ *p* < 0.05 vs. the S group; ** *p* < 0.01 and * *p* < 0.05 vs. the N group; ^##^ *p* < 0.01 vs. the P group; S: sham, normal saline; N: normal saline + CA/CPR; P: moderate hyperkalemia P + CA/CPR; Q: moderate hyperkalemia Q + CA/CPR.

**Table 1 brainsci-13-01285-t001:** The baseline characteristics among all groups.

Baseline Characteristics	S (*n* = 15)	N (*n* = 15)	P (*n* = 15)	Q (*n* = 15)
BW (g) (I)	228 ± 8	231 ± 12	228 ± 15	228 ± 11
K^+^ (mmol/L)	3.71 ± 0.41	3.5 ± 0.47	3.51 ± 0.36	3.62 ± 0.50
HR (Beats/min)	402 ± 36	400 ± 27	422 ± 40	423 ± 39
MAP (mmHg)	101 ± 9	104 ± 7	103 ± 15	104 ± 12
P wave amplitude (mV)	0.21 ± 0.02	0.20 ± 0.01	0.20 ± 0.04	0.21 ± 0.02
QRS complex duration (ms)	23 ± 1.10	24 ± 1.0	22 ± 1.2	24 ± 1.0
BW (g) (ROSC 24 h)	227 ± 7	213 ± 12	212 ± 15	211 ± 12

S: sham, normal saline; N: normal saline + CA/CPR; P: moderate hyperkalemia P + CA/CPR; Q: moderate hyperkalemia Q + CA/CPR; BW(I): Initial body weight; HR: heart rates; MAP: mean arterial pressure; K^+^: serum potassium; BW (g) (ROSC 24 h): Body weight 24 h after restoration of spontaneous circulation in rats.

**Table 2 brainsci-13-01285-t002:** The duration of administered liquid, the total amount of KCl administered, serum (K^+^) levels, HR, and MAP before CA among all groups.

Pre-CA	S (*n* = 15)	N (*n* = 15)	P (*n* = 15)	Q (*n* = 15)
T (min)	10	10	8.9 ± 0.5	13.04 ± 1.04
M _(KCl)_ (mg)	0	0	43.58 ± 3.88	63.89 ± 6.66
K^+^ (mmol/L)	3.66 ± 0.78	3.85 ± 0.47	8.76 ± 0.45 ***	10.18 ± 0.30 ***^###^
HR (Beats/min)	406 ± 35	418 ± 35	425 ± 53	415 ± 49
MAP (mmHg)	99 ± 6	101 ± 6	98 ± 13	90 ± 8 **^&&^

^&&^ *p* < 0.01 vs. the S group; ** *p* < 0.01 and *** *p* < 0.001 vs. the N group; ^###^ *p* < 0.001 vs. the P group; S: sham, normal saline; N: normal saline + CA/CPR; P: moderate hyperkalemia P + CA/CPR; Q: moderate hyperkalemia Q + CA/CPR; T: duration of administered liquid before CA; M _(KCl)_: total amount of KCl administered prior to CA; K^+^: serum potassium levels; HR: heart rates; MAP: mean arterial pressure.

## Data Availability

The data and analyses used in this study can be obtained from the corresponding author with a reasonable request.

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
