# Peer review of "Moderate Hyperkalemia Regulates Autophagy to Reduce Cerebral Ischemia-Reperfusion Injury in a CA/CPR Rat Model"

_brainsci, 2023, doi:10.3390/brainsci13091285_

Round 1

Reviewer 1 Report

Comments and Suggestions for Authors

The authors showed that the neuroprotective effect of hyperkalemia through the regulation of autophagy in CA/CPR rat model. Authors used CA/CPR model in rat for cerebral ischemia-reperfusion injury and the biochemical assay to observe the pathological changes. And autophagy signaling was investigated using the immunohistochemisty and western blot analysis.

Authors showed that the increased serum (K+) level after CA/CPR has a neuroprotective effect and this effect is mediated by the regulation of antophagy signaling. It is important to find the mechanism of neuroprotection by the increased serum (K+) level in CA/CPR animal model. This reviewer has some concerns about this manuscript.

1.       CA/CPR model has some variation between the animals. Please mention how many rats used in total and how many rats are used in analysis.

2.       This reviewer wonders that this model show the consistency in the pattern of neuronal death in whole cortex. Is there any possibility to choose with bias when authors select the region? Authors chose the cortex area for the analysis of neuronal death, which region of cortex and which layer was selected?

3.        In method, what size and which part of cortex was used for biochemical assay?

4.       In method of neurological evaluation, please describe more detail in numbering the evaluation.

5.       Authors showed the western blot data in Fig 5 and 7. Please show the whole blot with the size markers representing the data in the supplement figure.   

6.       In Fig 5, authors described the expression level of lysosomes in cortical neurons. So authors need to do the double immunohistochemistry with anti-NeuN, a neuronal marker to show that LC3 was changed in neurons.

Reviewer 2 Report

Comments and Suggestions for Authors

This preclinical study aimed to find the optimal degree of raising serum (K+) and to investigate the relationship between high (K+) and autophagy and the underlying mechanisms in a cardiac arrest/cardiopulmonary resuscitation (CA/CPR) rat model. This study found that raising serum (K+) levels led to decreased neurological dysfunction, improved neuronal morphology, reduced formation of autophagosomes and lysosomes, and lowered the LC3â…¡/LC3â…  ratio in CA/CPR rats 24 hours after resuscitation. These effects were attributed to the regulation of autophagy levels following cerebral ischemia-reperfusion through the mTOR-ULK1- Beclin1 signaling pathway. This research paper is well written based on appropriate methods and results. Here are my suggestions:

1.       Please present the sample size calculation.

2.       Please describe in detail how to randomly assign rats.

3.       Please clarify the inclusion/exclusion of the study rat by presenting the flow chart.

4.       How good was the intra- and inter-observer agreement for the neurological evaluation?

5.       Was there any difference in weight loss between groups after 24 hours of ROSC? Weight loss may affect neurological assessments and various molecular markers. Therefore, Table 1 should include the weight at the time of evaluation.

6.       What is the magnification of the tissue in Figures 3 and 4? Please describe in the table footnotes.

7.       Please provide the data used in the bar graph as supplementary tables.

Reviewer 3 Report

Comments and Suggestions for Authors

The group of authors showed that elevated levels of serum potassium confer greater protection against cerebral ischemia-reperfusion injury (CIRI) through the mTOR-ULK1-Beclin1 pathway, which efficiently modulates autophagic activity. The topic can be considered original and the defined involved mechanism is the novelty of the work compared to the similar published material. The number of references is appropriate; however, some improvements are mentioned in the following comments. In line with the evidence, a more detailed conclusion should be provided as well.

Other comments:

-Specific improvement is required to compare KCl with other protective agents in CIRI, such as hyperbaric oxygen.

-At the start of section 3.1, a short introductory paragraph is required.

-The resolution of Figure 5a should be increased.
